# Preoperative Tunnel Measurement in Hidradenitis Suppurativa: Comparison of Palpation and Ultrasound

**DOI:** 10.3390/diagnostics15111442

**Published:** 2025-06-05

**Authors:** Aslı Tatlıparmak, Murat Doğan, Zafer Türkoğlu

**Affiliations:** 1Department of Dermatology, Üsküdar University, 34768 İstanbul, Türkiye; 2Department of Dermatology, Başakşehir Çam and Sakura City Hospital, Health Sciences University, 34480 İstanbul, Türkiye; muratdogan1984@gmail.com (M.D.); cemzalu@gmail.com (Z.T.)

**Keywords:** tunnel length, hidradenitis suppurativa, surgical planning, ultrasound, palpation

## Abstract

**Background/Objectives**: The accurate measurement of tunnel lengths in hidradenitis suppurativa (HS) is critical for surgical planning. This study aimed to evaluate the agreement between palpation and high-frequency ultrasound (USG) for assessing tunnel lengths in HS patients. **Methods**: This prospective study included patients who underwent the surgical excision of tunnels between May 2024 and July 2024 at a referral dermatology clinic. Tunnel lengths were measured preoperatively using palpation and USG. Clinical and demographic data, including lesion localization and disease severity, were prospectively recorded and analyzed. **Results**: This study analyzed 121 lesions from patients undergoing surgical excision for HS. Tunnel lengths measured by palpation had a median of 30 mm [IQR 18–40], while USG measurements had a median of 36 mm [IQR 24–51.5], with USG identifying tunnels 10.3 mm longer on average (95% CI: 8.2–12.3). Axillary lesions were most frequent (53.7%), followed by inguinal (32.2%) and sacral regions (6.6%). Most lesions were classified as Hurley stage 2 (59.5%) and stage 3 (37.2%), with a median IHS4 score of 8 [IQR 7–11]. **Conclusions**: High-frequency USG offers greater precision than palpation in measuring tunnel lengths, indicating its potential to enhance disease assessments in HS.

## 1. Introduction

Hidradenitis suppurativa (HS) is a chronic inflammatory skin disease characterized by painful nodules, abscesses, and sinus tracts, commonly referred to as tunnels [1]. These tunnels form due to chronic inflammation and repeated abscess formation, where ongoing tissue destruction creates interconnected subcutaneous tracts [2]. Typically emerging in advanced stages of HS, tunnels are critical indicators of disease severity and often necessitate surgical intervention, as they represent a chronic disease that is unresponsive to medical therapy alone [3,4]. Lined with granulation tissue or epithelium, tunnels are resistant to antibiotics and biologics, stressing the importance of complete surgical removal to prevent recurrence [5]. However, surgical planning requires a careful assessment, as excessive excision may lead to increased morbidity and impaired healing [6]. Accurately determining the tunnel length and extent is crucial for staging the disease, guiding surgical interventions, and optimizing outcomes [7].

Palpation, based on manual examinations, is commonly used in clinical settings to detect the presence and estimate the extent of tunnels in HS. While it is not a standardized or validated method for quantifying tunnel length, it remains a practical tool in routine dermatologic assessments. However, palpation is inherently subjective and may fail to detect deeper, branching, or subclinical tunnels. In contrast, high-frequency ultrasound (USG) has emerged as a promising imaging modality that enables the detailed visualization of dermal and subcutaneous structures [8]. USG can identify tunnel complexity and extent with greater precision, offering potential advantages for both diagnosis and surgical planning [9]. However, the comparative accuracy and reliability of USG and palpation in measuring tunnel lengths remain unexplored.

Despite the growing use of high-frequency ultrasound (USG) in dermatology, its specific application in measuring tunnel lengths in hidradenitis suppurativa (HS) remains underexplored. Palpation, though commonly used in clinical practice for tunnel detection, has notable limitations in identifying deeper or subclinical extensions. This highlights the need for more objective and reproducible assessment tools. Demonstrating the accuracy and reliability of USG in this context could enhance diagnostic precision and support more informed surgical planning. This study aims to evaluate the agreement between palpation and USG in preoperative tunnel length assessment in HS patients. By addressing this gap, the findings seek to support evidence-based refinements in clinical evaluation and inform future research on outcome-based utility.

## 2. Materials and Methods

### 2.1. Study Design and Setting

This study was a method comparison analysis conducted prospectively at an outpatient dermatology clinic, which serves as a referral center for the surgical management of HS. The primary aim was to evaluate the agreement between palpation and high-frequency USG in assessing tunnel lengths in patients undergoing surgery for HS. Secondary objectives included identifying lesion localization, determining disease severity through established staging systems, and examining the clinical characteristics of the study population.

### 2.2. Study Population

Patients diagnosed with HS who underwent the surgical excision of tunnels between May 2024 and August 2024 were included. Surgical decisions were guided by established practices, emphasizing wide excision as the preferred treatment modality for chronic and advanced stages of HS, particularly in cases unresponsive to medical therapy [10]. Wide excision was chosen where applicable to ensure the removal of irreversibly damaged tissue, including tunnels and scars, with the goal of optimizing disease control and minimizing recurrence risk. Eligible patients were those with a confirmed diagnosis of HS, persistent tunnels requiring surgical intervention, and complete preoperative data. Persistent tunnels were defined as non-healing lesions associated with recurrent symptoms or ongoing inflammation. Patients with incomplete medical records, coexisting dermatological conditions affecting subcutaneous tissue, or a history of surgery at the same site were excluded to ensure the accuracy of tunnel measurement and clinical outcomes.

### 2.3. Tunnel Length Measurement Methods

Tunnel length measurements were obtained preoperatively using palpation and high-frequency ultrasound with a 20 MHz linear probe. Palpation was conducted by two experienced dermatologists independently through a manual examination to estimate the extent of tunnels based on tactile feedback. Both dermatologists were blinded to each other’s measurements and to the USG results. Measurements were recorded in millimeters, and the average of the two dermatologists’ assessments was taken as the final value.

For USG imaging, if the tunnel’s start and end were visible in a single ultrasound image, measurements were taken directly using a digital caliper integrated into the device. When the tunnel could not be visualized in a single image, the ultrasound probe was positioned at a 90-degree angle to the skin surface, and the start and end points of the tunnel were marked on the skin. The distance between these points was then measured over the skin using a measuring tape. The USG identified the full extent of tunnels, including subclinical branches and surrounding inflammation. The final value for USG measurements was calculated as the average of the two dermatologists’ assessments, who were blinded to the palpation results.

### 2.4. Data Collection for Secondary Objectives

Demographic and clinical data were prospectively recorded to support the secondary objectives. Collected information included age, sex, body mass index (BMI), smoking status, and family history of HS. Disease severity was assessed using the Hurley staging system, which classifies HS into stages 1, 2, and 3 based on the presence and extent of abscesses, nodules, and tunnels [11]. Additional assessments included the International Hidradenitis Suppurativa Severity Score System (IHS4) and the Severity of Skin in Hidradenitis Suppurativa (SOS-HS) staging [12,13]. Treatment history, including the use of antibiotics, isotretinoin, and biologics like adalimumab, as well as treatment durations, was also documented. Lesion localization was categorized by the anatomical site, including axilla, inguinal, sacral, gluteal, areolar, perineal, pubic, and sternal regions.

### 2.5. Ethical Approval

Ethical approval was obtained from the Üsküdar University Non-Interventional Research Ethics Committee (date: 26 April 2024, no: 61351342/020-15). Written informed consent was obtained from all patients prior to their inclusion in the study. All data were anonymized to ensure confidentiality and compliance with ethical standards. This study was conducted in accordance with the ethical principles outlined in the Declaration of Helsinki.

### 2.6. Sample Size

The sample size was calculated to require a minimum of 50 patients, based on recommendations for method comparison studies [14]. Additionally, considering the estimated prevalence of HS at approximately 0.7–1.2% in the general population, and that around one-third to one-half of patients present with moderate to severe disease (Hurley stage II/III) requiring surgical intervention, the recruitment of 50 patients was deemed both feasible and clinically representative [15,16]. Ultimately, 51 patients were enrolled in this study.

### 2.7. Analysis

The statistical analysis was conducted using Statistical Package for the Social Sciences (SPSS) version 30.0 (IBM Corp., Armonk, NY, USA) and MedCalc Statistical Software (version 20.104). Descriptive statistics for continuous variables were expressed as mean ± standard deviation (SD) or median (interquartile range [IQR]), and categorical variables were presented as frequencies and percentages. Normality was assessed using histograms. To assess inter-rater agreement between the two dermatologists performing palpation and USG measurements, the intraclass correlation coefficient (ICC) was calculated using a two-way random-effects model with absolute agreement. Both single and average measures of the ICC were reported to provide a comprehensive evaluation of agreement. The ICC analysis ensured the reliability of both methods and validated the consistency of the measurements obtained independently by the dermatologists. A combined approach using Bland–Altman and Passing–Bablok regression analyses was employed to ensure a robust evaluation of agreement and to identify systematic and proportional biases between measurement methods [17]. The agreement between palpation and USG measurements was evaluated using Bland–Altman and Passing–Bablok regression analyses. The Bland–Altman analysis was used to calculate the mean bias and limits of agreement (LoAs) to assess systematic differences and variability between the methods. LoAs were defined as the mean bias ± 1.96 standard deviations of the differences [18]. The assumption of a normal distribution for measurement differences was verified prior to performing the Bland–Altman analysis. Passing–Bablok regression was used to evaluate systematic and proportional biases between palpation and USG measurements. The intercept and slope were calculated, with a slope value including 1 within the 95% confidence interval (CI) indicating no proportional bias, and an intercept including 0 within the 95% CI suggesting no systematic bias [19]. The residual standard deviation (RSD) and ±1.96 RSD intervals were also reported to assess variability. The strength of the relationship between the two methods was determined using Spearman rank correlation. Statistical significance was defined as *p* < 0.05, and all statistical tests were two-tailed.

## 3. Results

This study included a total of 121 lesions from 51 patients. The mean age of the patients was 32 ± 9.4 years, and 29 (56.9%) were female. A family history of HS was reported in 20 patients (39.2%), and the median time since diagnosis was 48 (IQR 24–96) months. Smoking was reported in 31 patients (60.8%), with a median BMI of 22.5 (IQR 21–25.4). Antibiotic use was documented in 37 patients (72.5%), isotretinoin use in 24 (47.1%), and adalimumab use in 15 (29.4%), with a mean treatment duration of 26 ± 16 months. A total of twenty-four patients (47.1%) were treated with ≥2 therapies, while only one (2%) received ≥3 therapies (Table 1).

Among the 121 lesions analyzed, the axilla was the most common site (*n* = 65, 53.7%), followed by the inguinal region (*n* = 39, 32.2%). Other sites included sacral (*n* = 8, 6.6%), gluteal (*n* = 5, 4.1%), and less frequent locations such as areolar, perineal, pubic, and sternal regions (each *n* = 1, 0.8%). Based on the Hurley staging system, stage 2 lesions were most prevalent (*n* = 72, 59.5%), followed by stage 3 (*n* = 45, 37.2%) and stage 1 (*n* = 4, 3.3%). The median IHS4 was 8 (IQR 7–11), while the SOS-HS staging revealed that most lesions were stage 3 (*n* = 74, 61.2%), with stage 2 and stage 1 accounting for forty-six (38.0%) and one (0.8%) lesions, respectively. Tunnel lengths measured by USG were 36 (IQR 24–51.5) mm, and those measured by palpation were 30 (IQR 18–40) mm (Table 2). Representative clinical and sonographic images are shown in Figure 1.

Lesion sites were further stratified by Hurley stage to assess the anatomical variation in disease severity (Table 3). Axillary and inguinal lesions were mostly categorized as stage 2 and 3. Gluteal lesions were predominantly stage 3 (75.0%), while sacral lesions were primarily stage 2 (88.9%). Lesions in perineal, pubic, and sternal regions were classified as advanced (stage 2 or 3), with no stage 1 cases observed.

The agreement between the two dermatologists for palpation measurements was evaluated using the ICC. The single measures ICC for palpation was 0.928 (95% CI: 0.314–0.978), indicating good agreement, and the average measures ICC was 0.963 (95% CI: 0.478–0.989), demonstrating excellent reliability. Similarly, the agreement between dermatologists for USG measurements showed excellent reliability, with a single measures ICC of 0.991 (95% CI: 0.640–0.998) and an average measures ICC of 0.996 (95% CI: 0.780–0.999).

During the 3-month follow-up period, no recurrences were observed. As a surgical complication, suture dehiscence was reported in two patients.

The Bland–Altman analysis showed a mean bias of −10.3 mm (95% CI: −12.3 to −8.2, *p* < 0.001) between palpation and USG measurements, with LoA ranging from −32.7 (95% CI: −36.3 to −29.2) to 12.2 mm (95% CI: 8.6 to 15.7) (Table 4, Figure 2).

The Passing–Bablok regression analysis revealed a systematic bias with an intercept of −3.2 (95% CI: −5.0 to −1.67) and a proportional bias with a slope of 0.88 (95% CI: 0.83 to 0.94). The regression equation was calculated as *y* = −3.23 + 0.88*x*, where x represents the tunnel length measured by USG (mm) and y represents the tunnel length measured by palpation (mm). The RSD was 6.3 (±1.96 RSD interval: −12.4 to 12.4). The Cusum test confirmed no significant deviation from linearity (*p* = 0.65). The Spearman rank correlation coefficient was 0.919 (95% CI: 0.886 to 0.943, *p* < 0.0001) (Table 5, Figure 3).

## 4. Discussion

This study evaluated the agreement between palpation and USG in measuring tunnel lengths in patients with HS scheduled for surgery performed by dermatologists. Preoperative measurements revealed a general correlation between the two methods, indicating that palpation can provide a basic estimation of tunnel extent. However, there was notable variability, with USG often identifying longer tunnels than palpation. This variability highlights the limitations of palpation in fully capturing tunnel involvement, particularly subclinical extensions.

In HS, USG has emerged as a pivotal tool for understanding the disease’s underlying mechanisms and aiding in its clinical management [20]. High-frequency USG, as demonstrated by Wortsman et al., can detect early pathological changes such as patterns of keratin fragmentation, fluid collections, and the initial stages of tunnel formation, which are often missed during palpation or clinical examination [21]. Similarly, Gogate et al.’s pilot study highlighted the utility of USG and Doppler ultrasound in accurately identifying disease extent and activity in HS patients, often uncovering subclinical lesions and refining disease staging [22]. Notably, their findings revealed that USG not only identified misdiagnosed cases but also altered the management plan in 26% of patients, emphasizing its role in distinguishing between medical and surgical candidates. These studies collectively underline the critical role of USG in visualizing subclinical changes and tailoring patient-specific interventions. Building on these foundational insights, the agreement analyses in this study further emphasize the limitations of palpation and the advantages of USG in assessing tunnel lengths. The Bland–Altman analysis revealed a mean bias of −10.3 mm (95% CI: −12.3 to −8.2), demonstrating that palpation consistently underestimates tunnel lengths compared to USG. This systematic underestimation has significant clinical implications, as the incomplete identification of tunnel extent can lead to insufficient surgical excision and an increased risk of recurrence. The wide LoAs, ranging from −32.7 to 12.2 mm, stress the variability of palpation, reinforcing its limitations as a standalone method. This aligns with findings from other medical contexts, where USG has proven superior to palpation in applications such as inter-rectus muscle measurement, intervertebral level determination, splenomegaly detection, and glenohumeral subluxation assessments [23,24,25,26]. Similarly, the Passing–Bablok regression analysis revealed systematic and proportional biases, with an intercept of −3.2 (95% CI: −5.0 to −1.67) and a slope of 0.88 (95% CI: 0.83 to 0.94), suggesting that the underestimation by palpation increases with tunnel length. These findings validate the role of USG in surgical planning, where a precise assessment is crucial to optimizing outcomes and minimizing recurrence risk.

Recent clinical investigations have reinforced the potential role of USG in optimizing surgical outcomes in HS. In our study, palpation consistently underestimated tunnel lengths compared to high-frequency USG, raising the possibility that subclinical extensions may be missed, potentially affecting surgical completeness. Supporting this concern, Michelucci et al. demonstrated that preoperative mapping with ultra-high-frequency ultrasound followed by wide local excision led to a 90% remission rate over 22 months, suggesting that improved margin definition may reduce recurrence [27]. Similarly, Rao et al. found that high-resolution USG and color Doppler ultrasound frequently upstaged clinical disease severity, uncovering a more extensive involvement in over half of the patients [28]. While our study focused on diagnostic agreement rather than outcomes, these collective findings highlight the clinical relevance of accurate preoperative tunnel assessments. Future studies should investigate whether the use of USG directly improves surgical endpoints such as recurrence, complication rates, and healing times.

HS is a profoundly debilitating disease, as evidenced by the extended diagnostic delays and complex therapeutic trajectories observed in this study. The median time since diagnosis was 48 months, a relatively encouraging finding compared to the 7.23 ± 2.81 years reported in a pooled analysis of 9286 patients [29]. While effective therapies, including biologics, are available, the long disease course and variability in patient response complicate management. For instance, despite antibiotics being a first-line therapy, 30% of patients in our study were not taking them at the time, reflecting challenges such as discontinuation, side effects, or the need to escalate to advanced therapies [10]. Biologics were used by 29.4% of patients, indicating their increasing adoption in advanced disease. However, the continued need for surgical intervention highlights the limitations of pharmacological options. Even with promising results from PIONEER I (41.8%) and PIONEER II (58.9%) regarding HS Clinical Response at 12 weeks, many patients remain in search of a definitive treatment, suggesting biologics are not a universal solution [30,31]. Approximately 50% of patients required multiple therapies, and given that this cohort comprised surgical cases, the findings highlight the need for individualized strategies and continued research to optimize both medical and surgical approaches.

Lesion localization plays a critical role in the management of HS, influencing surgical planning and postoperative outcomes [6]. In our study, the axilla was the most commonly affected site, aligning with the existing literature [32]. This localization presents unique challenges due to its anatomical complexity and proximity to important structures. Hladiuk et al. demonstrated that axillary surgery can significantly impact arm mobility, emphasizing the need for careful planning [33]. Surgeons must balance wide excision to minimize recurrence with preserving function to avoid contractures or impairment. In such scenarios, a precise preoperative assessment becomes crucial, and USG demonstrates significant value. By accurately visualizing subclinical tunnels and their branching patterns, USG helps define appropriate margins and supports planning excisions that are both effective and functional.

In addition to lesion localization, disease severity staging and treatment planning often rely on established scoring systems such as the Hurley classification, IHS4, and SOS-HS staging. These tools provide a valuable framework for understanding the extent and severity of HS, as reflected in our study, where the majority of lesions were categorized as Hurley stage 2 (59.5%) and stage 3 (37.2%). With a median IHS4 score of 8 (IQR 7–11) and SOS-HS staging showing 61.2% of lesions in stage 3, these scores underscore the significant disease burden faced by patients. Moreover, newer systems, like the Hidradenitis Suppurativa Clinical Response (HiSCR), which classifies patients based on a 50% reduction in abscesses and inflammatory nodules, continue to shape HS assessment, although their utility in daily clinical practice remains uncertain [34]. Despite their value, these systems may fail to capture subclinical or complex anatomical features critical for guiding treatment decisions [35]. In our cohort of surgical candidates, USG addressed these limitations by refining staging and identifying subclinical tunnels that might have been overlooked by traditional scoring. By integrating USG with clinical scoring, clinicians can adopt a more nuanced and precise approach to disease evaluation, ensuring that both medical and surgical strategies are optimized to meet individual patient needs. These findings emphasize the evolving role of imaging modalities in enhancing the effectiveness of HS management frameworks.

### Limitations

This study has several limitations that should be considered when interpreting the findings. First, although the dermatologists performing palpation and USG measurements were blinded to each other’s results, variability in operator experience and technique might have influenced the measurements. While excellent inter-rater reliability was demonstrated, differences inherent to manual and imaging techniques could still exist. Second, the study population was drawn from a single surgical referral center, which may limit the generalizability of the findings. Patients referred for surgery often represent more advanced or treatment-resistant cases, and these results may not reflect the broader spectrum of HS seen in primary care or general dermatology settings. Moreover, being a single-center study may limit variation in practice patterns, physician experience, ultrasound interpretation techniques, and patient demographics, all of which could influence tunnel detection and measurement. Third, while USG showed superior precision in assessing tunnel lengths, its reliance on operator expertise and the availability of high-frequency devices could pose challenges for wider implementation in routine clinical practice. In particular, many dermatology clinics may lack access to high-resolution ultrasound equipment or trained personnel, limiting the feasibility of adopting USG as a routine diagnostic tool in all settings. Lastly, the follow-up period was limited to three months, which, while not the primary focus of this study, may restrict the assessment of long-term outcomes such as recurrence rates and postoperative complications. Future studies with longer follow-up periods, larger multicenter cohorts, and standardized protocols are warranted to confirm these findings and evaluate their applicability in diverse clinical settings.

## 5. Conclusions

This study revealed key differences between palpation and USG in measuring tunnel lengths in HS. Palpation tended to measure shorter tunnel lengths compared to USG, highlighting its limitations in detecting deeper or more complex tunnels. These findings suggest that USG may offer valuable additional detail that could support more informed surgical planning. However, to establish its impact on long-term outcomes, future prospective studies comparing imaging-based assessments with clinical endpoints are warranted.

## Figures and Tables

**Figure 1 diagnostics-15-01442-f001:**
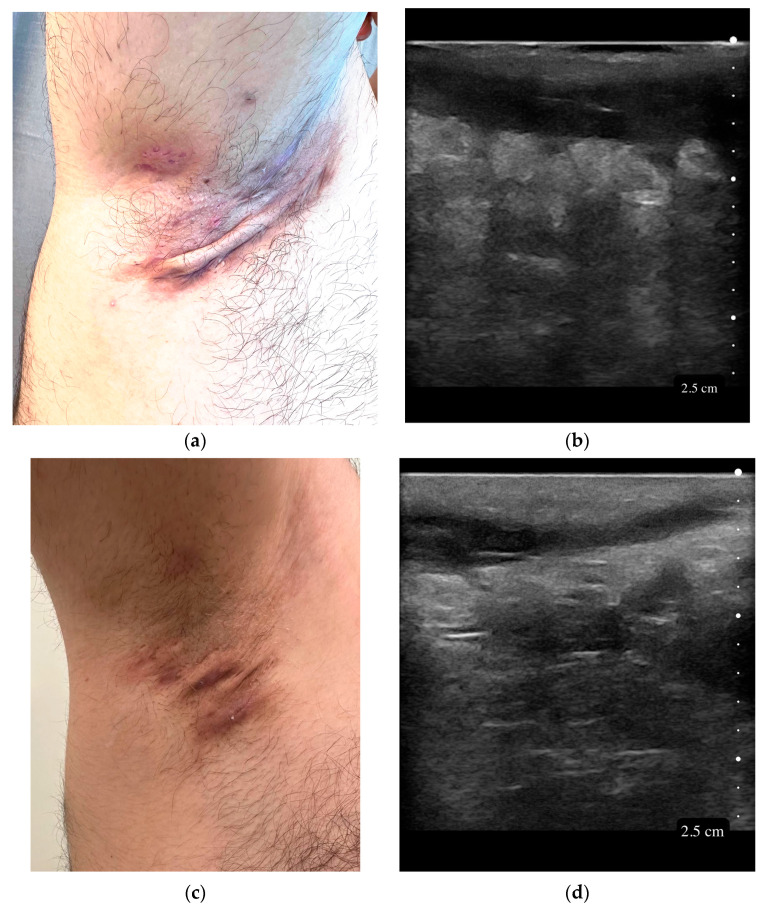
Clinical and sonographic images of axillary tunnels in hidradenitis suppurativa. (**a**) Clinical image showing a single, wide, linear axillary tunnel with superficial tract formation. (**b**) Corresponding ultrasound image demonstrating a straight, hypoechoic tunnel in the subcutaneous tissue; the tunnel length was similarly estimated by palpation and ultrasound. (**c**) Clinical image showing a narrow, fibrotic axillary tunnel with subtle skin depression and no overt sinus tract. (**d**) Corresponding ultrasound image revealing a deeper, hypoechoic tunnel extending beyond the clinical margins, showing greater extension than clinically appreciable by palpation. High-frequency ultrasound images were obtained using a 20 MHz linear probe (Clarius L20 HD3, Clarius Mobile Health Corp., Vancouver, BC, Canada).

**Figure 2 diagnostics-15-01442-f002:**
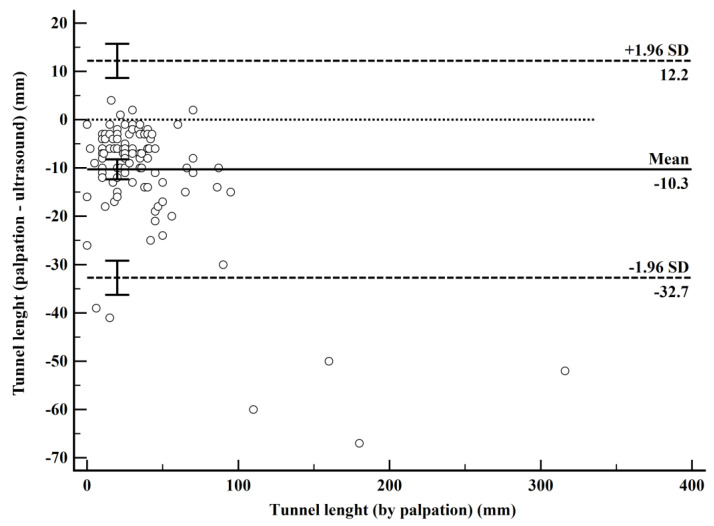
Bland–Altman plot comparing tunnel length measurements by palpation and ultrasound.

**Figure 3 diagnostics-15-01442-f003:**
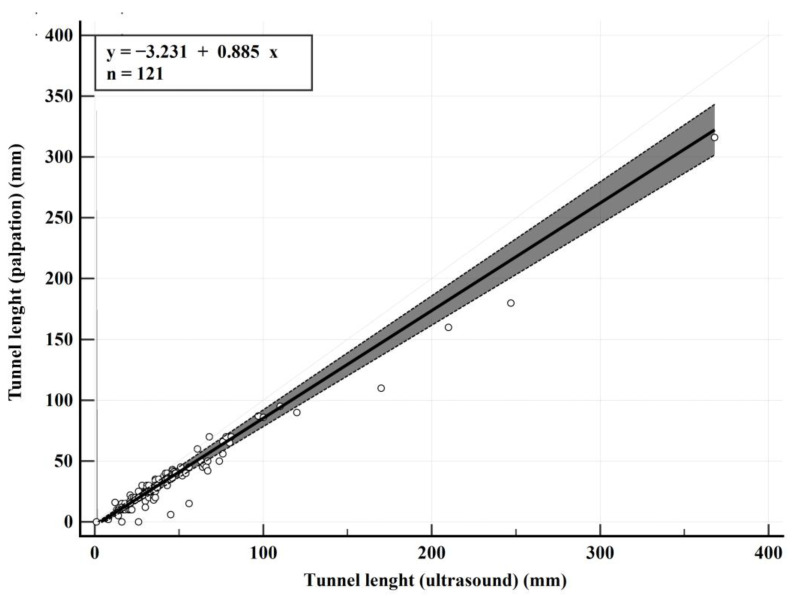
Passing–Bablok regression plot comparing tunnel length measurements by palpation and ultrasound.

**Table 1 diagnostics-15-01442-t001:** Baseline characteristics of patients.

Variable	All Patients (*n* = 51)
Age	32 ± 9.4
Sex (female)	29 (56.9%)
Family history	20 (39.2%)
Time since diagnosis (months)	48 [24–96]
Smoking	31 (60.8%)
BMI	22.5 [21–25.4]
Use of antibiotics	37 (72.5%)
Use of isotretinoin	24 (47.1%)
Use of adalimumab	15 (29.4%)
Duration of adalimumab treatment (months)	26 ± 16
Treated with ≥2 therapies	24 (47.1%)
Treated with ≥3 therapies	1 (2%)

(Data presented as mean ± SD, median [IQR], or *n* (%)); BMI: body mass index.

**Table 2 diagnostics-15-01442-t002:** Characteristics of lesions.

Variable	Category	Lesions (*n* = 121)
Lesion Site	Axilla	65 (53.7%)
	Inguinal	39 (32.2%)
	Sacral	8 (6.6%)
	Gluteal	5 (4.1%)
	Areolar	1 (0.8%)
	Perineal	1 (0.8%)
	Pubic	1 (0.8%)
	Sternal	1 (0.8%)
Hurley Stage	Stage 1	4 (3.3%)
	Stage 2	72 (59.5%)
	Stage 3	45 (37.2%)
IHS4	Score	8 [7–11]
SOS-HS Stage	Stage 1	1 (0.8%)
	Stage 2	46 (38.0%)
	Stage 3	74 (61.2%)
Tunnel Length (mm)	By ultrasound	36 [24–51.5]
	By palpation	30 [18–40]

(Data presented as mean ± SD, median [IQR], or *n* (%)); IHS4: International Hidradenitis Suppurativa Severity Score; SOS-HS: Severity of Skin in Hidradenitis Suppurativa.

**Table 3 diagnostics-15-01442-t003:** Lesion site stratified by Hurley stage.

Lesion Site	Hurley Stage 1	Hurley Stage 2	Hurley Stage 3
Axilla	1.6% (1)	61.9% (39)	36.5% (23)
Gluteal	0.0% (0)	25.0% (1)	75.0% (3)
Inguinal	5.0% (2)	55.0% (22)	40.0% (16)
Other	50.0% (1)	50.0% (1)	0.0% (0)
Perineal	0.0% (0)	0.0% (0)	100.0% (1)
Pubic	0.0% (0)	0.0% (0)	100.0% (1)
Sacral	0.0% (0)	88.9% (8)	11.1% (1)
Sternal	0.0% (0)	100.0% (1)	0.0% (0)
Total	3.3% (4)	59.5% (72)	37.2% (45)

Data presented as % (*n*); row percentages shown.

**Table 4 diagnostics-15-01442-t004:** Bland–Altman analysis of tunnel length measurements by palpation and ultrasound.

Comparison	Mean Bias (95% CI)	*p*	Lower LoA (95% CI)	Upper LoA (95% CI)
Palpation-USG	−10.3	<0.001	−32.7 (−36.3 to −29.2)	12.2 (8.6 to 15.7)

USG: ultrasonography, CI: confidence interval, LoA: limits of agreement.

**Table 5 diagnostics-15-01442-t005:** Passing–Bablok regression analysis of tunnel length measurements by palpation and ultrasound.

Comparison	Intercept A (95% CI)	Slope B (95% CI)	RSD (±1.96 SD)
Palpation-USG	−3.2 (−5 to −1.67)	0.88 (0.83 to 0.94)	6.3 (−12.4 to 12.4)

USG: ultrasonography, CI: confidence interval, RSD: residual standard deviation, SD: standard deviation.

## Data Availability

The data that support the findings of this study are available from the corresponding author upon reasonable request.

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
