# Peer review of "Preoperative Tunnel Measurement in Hidradenitis Suppurativa: Comparison of Palpation and Ultrasound"

_diagnostics, 2025, doi:10.3390/diagnostics15111442_

Round 1
Reviewer 1 Report
Comments and Suggestions for Authors
This is a well-structured, methodologically sound, and clinically relevant article that investigates the agreement between palpation and high-frequency ultrasound in the preoperative measurement of tunnels in hidradenitis suppurativa. The topic is timely and contributes to the growing evidence supporting the role of imaging in HS surgical planning. However the manuscript need minor formal and structural corrections.
- Study Population Clarity: The number of patients vs. lesions is clearly reported, but the anatomical distribution could benefit from stratification by Hurley stage.
- Clarify the HFUS frequencies used for the examination
- External Validity: The article should discuss more explicitly the limitations in generalizability due to being a single-center study in a surgical referral setting.
- While the discussion references relevant literature, some claims could be better substantiated, e.g., the role of HFUS in modifying surgical plans and reducing the risk of recurrence should be supported by quantitative evidence or examples. I give you some suggestions: (Michelucci A, Fidanzi C, Manzo Margiotta F, et al. Presurgical Mapping With Ultra-high Frequency Ultrasound of Hidradenitis Suppurativa Lesions Treated With Wide Local Excision and Secondary Intention Healing. Dermatol Surg. 2025;51(1):36-39. doi:10.1097/DSS.0000000000004349; Rao S, Sharma A, Kumaran MS, Narang T, Sinha A, Dogra S. Disease severity assessment in hidradenitis suppurativa: A single-centre cross-sectional study utilising clinical evaluation, high-resolution ultrasound and colour doppler. Indian J Dermatol Venereol Leprol. 2025;91(2):145-151. doi:10.25259/IJDVL_542_2024..)
-
Including outcome data beyond tunnel length measurement—such as recurrence rates, complication rates, or postoperative healing times—in patients assessed with HFUS versus palpation. Alternatively, if such data are not available, the authors could temper their conclusion by clearly stating that while USG shows superior diagnostic accuracy, the direct impact on surgical outcomes in their cohort was not evaluated and warrants future prospective investigation.
Author Response
|
Comments 1: This is a well-structured, methodologically sound, and clinically relevant article that investigates the agreement between palpation and high-frequency ultrasound in the preoperative measurement of tunnels in hidradenitis suppurativa. The topic is timely and contributes to the growing evidence supporting the role of imaging in HS surgical planning. However the manuscript need minor formal and structural corrections. |
|
Response 1: We sincerely thank the reviewer for their positive and encouraging evaluation of our manuscript. We greatly appreciate the recognition of the study’s methodological rigor and clinical relevance, as well as the constructive suggestions provided. We have addressed all the points raised and revised the manuscript accordingly to improve its clarity and structure.
Comments 2: Study Population Clarity: The number of patients vs. lesions is clearly reported, but the anatomical distribution could benefit from stratification by Hurley stage. Response 2: We appreciate the reviewer’s thoughtful suggestion. In response, we have added a new table (now Table 3) that presents lesion site distribution stratified by Hurley stage.
Comments 3: Clarify the HFUS frequencies used for the examination Response 3: We thank the reviewer for this important clarification request. We have now specified that all ultrasound examinations were performed using a high-frequency 20 MHz linear probe. This information has been added to the Methods section under “Tunnel Length Measurement Methods” to improve methodological clarity.
Comments 4: External Validity: The article should discuss more explicitly the limitations in generalizability due to being a single-center study in a surgical referral setting. Response 4: We thank the reviewer for this important observation. We have revised the Limitations section to more clearly emphasize that the single-center, surgical referral nature of the study may limit generalizability—not only due to disease severity but also because of limited variability in clinical practices, physician experience, and patient demographics.
Comments 5: While the discussion references relevant literature, some claims could be better substantiated, e.g., the role of HFUS in modifying surgical plans and reducing the risk of recurrence should be supported by quantitative evidence or examples. I give you some suggestions: (Michelucci A, Fidanzi C, Manzo Margiotta F, et al. Presurgical Mapping With Ultra-high Frequency Ultrasound of Hidradenitis Suppurativa Lesions Treated With Wide Local Excision and Secondary Intention Healing. Dermatol Surg. 2025;51(1):36-39. doi:10.1097/DSS.0000000000004349; Rao S, Sharma A, Kumaran MS, Narang T, Sinha A, Dogra S. Disease severity assessment in hidradenitis suppurativa: A single-centre cross-sectional study utilising clinical evaluation, high-resolution ultrasound and colour doppler. Indian J Dermatol Venereol Leprol. 2025;91(2):145-151. doi:10.25259/IJDVL_542_2024..) Including outcome data beyond tunnel length measurement—such as recurrence rates, complication rates, or postoperative healing times—in patients assessed with HFUS versus palpation. Alternatively, if such data are not available, the authors could temper their conclusion by clearly stating that while USG shows superior diagnostic accuracy, the direct impact on surgical outcomes in their cohort was not evaluated and warrants future prospective investigation. Response 5: We appreciate the reviewer’s thoughtful suggestion. In response, we have revised the discussion section to cite the studies by Michelucci et al. (2025) and Rao et al. (2025), which support the role of HFUS in modifying surgical plans and improving disease staging. Additionally, we have clarified that while our findings demonstrate superior diagnostic accuracy of USG over palpation in tunnel length measurement, the impact on surgical outcomes was not evaluated and warrants future investigation. Furthermore, we have revised the final paragraph of the discussion and the conclusion of the abstract to better reflect the scope of our data and avoid overinterpretation regarding surgical planning. Thank you for helping us improve the clarity and accuracy of our conclusions.
|

Reviewer 2 Report
Comments and Suggestions for Authors
Overall, the authors tried to evaluate the agreement between palpation and high-frequency ultrasound for assessing tunnel lengths in HS patients. Although the methods and materials to conduct the study were described detailed in part 2, definition of tunnels, indication for tunnel draining, surgical options had not been mentioned clearly. Moreover, there was no consensus recommendation as well as standard guidelines for surgical excision in hidradenitis suppurativa so that using palpation palpation as a preoperative tunnel measurement tool might be not appropriate.
Here are some more comments need to be clarify:
- The reason why choosing palpation as a measure tool to compare with ultrasound had not been explained because palpation was also not a standard tunnel measurement.
- Line 19: What was the abbreviation USG stand for? Abbreviation should not be in the abstract due to difficulty in understanding.
- Line 38: overly and aggressive used at the same time was quite irrelevant
- The time of study was inconsistent between the abstract and the materials and methods: which one is correct 2023 or 2024?
- Line 183: mistake correction: length not lenght
- One more limitation of this study was that most of dermatologic clinic might not have ultrasound to conduct this measure.
Author Response
|
Comments 1: Overall, the authors tried to evaluate the agreement between palpation and high-frequency ultrasound for assessing tunnel lengths in HS patients. Although the methods and materials to conduct the study were described detailed in part 2, definition of tunnels, indication for tunnel draining, surgical options had not been mentioned clearly. Moreover, there was no consensus recommendation as well as standard guidelines for surgical excision in hidradenitis suppurativa so that using palpation palpation as a preoperative tunnel measurement tool might be not appropriate. |
|
Response 1: Thank you for your thoughtful comment. We agree that there is currently no universally accepted gold standard for preoperative tunnel assessment in hidradenitis suppurativa. However, palpation remains widely used in routine clinical practice to detect the presence and approximate extent of tunnels, particularly in the absence of advanced imaging tools. As such, we chose palpation as the practical comparator for high-frequency ultrasound in this study. To clarify this point, we have revised the Introduction to better articulate the rationale for using palpation as a reference modality. We also acknowledge the lack of standardized guidelines for tunnel excision in HS. However, the primary aim of this study was to quantify agreement between two assessment methods, not to evaluate surgical outcomes. We have revised relevant sections to clarify these distinctions where appropriate. All changes are marked in the revised manuscript.
Comments 2: The reason why choosing palpation as a measure tool to compare with ultrasound had not been explained because palpation was also not a standard tunnel measurement. Response 2: Thank you for this important comment. We agree that palpation is not a validated or standardized tool for measuring tunnel length; however, it remains the most commonly used method in clinical practice to approximate tunnel presence and extent. We have now clarified this rationale in the Introduction, emphasizing that our aim was to assess how this widely used—but subjective—approach compares to high-frequency ultrasound in the preoperative evaluation of HS tunnels.
Comments 3: Line 19: What was the abbreviation USG stand for? Abbreviation should not be in the abstract due to difficulty in understanding. Response 3: Thank you for pointing this out. We have revised the abstract to spell out “ultrasound (USG)” at first mention and removed the abbreviation thereafter, in accordance with journal guidelines and to enhance clarity for all readers.
Comments 4: Line 38: overly and aggressive used at the same time was quite irrelevant Response 4: We appreciate the reviewer’s attention to language precision. The sentence has been revised for clarity and conciseness. “Overly aggressive” was replaced with “excessive” to avoid redundancy and improve the tone. Thank you for the helpful suggestion.
Comments 5: The time of study was inconsistent between the abstract and the materials and methods: which one is correct 2023 or 2024? Response 5: We thank the reviewer for pointing this out. The correct study period is 2024, as stated in the Materials and Methods section. We have corrected the date in the Abstract to ensure consistency across the manuscript.
Comments 6: Line 183: mistake correction: length not lenght Response 6: Thank you for noting this typographical error. We have corrected the spelling from “lenght” to “length” in the revised figures.
Comments 7: One more limitation of this study was that most of dermatologic clinic might not have ultrasound to conduct this measure. Response 7: Thank you for this valuable observation. We agree that access to high-frequency ultrasound devices and trained personnel may be limited in many dermatology clinics. We have now acknowledged this important limitation in the revised manuscript and expanded our limitations paragraph accordingly to reflect potential barriers to broader clinical implementation of USG-based assessments. |

Round 2
Reviewer 2 Report
Comments and Suggestions for Authors
Thank you for your explanation. I agree with the present form.
Author Response
Thank you very much